# LPS Administration Impacts Glial Immune Programs by Alternative Splicing

**DOI:** 10.3390/biom12020277

**Published:** 2022-02-08

**Authors:** Vladimir N. Babenko, Galina T. Shishkina, Dmitriy A. Lanshakov, Ekaterina V. Sukhareva, Nikolay N. Dygalo

**Affiliations:** Laboratory of Functional Neurogenomics, Federal Research Center Institute of Cytology and Genetics, Siberian Branch of the Russian Academy of Science, 630090 Novosibirsk, Russia; lanshakov@bionet.nsc.ru (D.A.L.); sev@bionet.nsc.ru (E.V.S.); dygalo@bionet.nsc.ru (N.N.D.)

**Keywords:** lipopolysaccharide, hippocampus, RNA-Seq, alternative splicing, immune response

## Abstract

We performed transcriptome analysis in the hippocampus 24 h after lipopolysaccharide (LPS) administration. We observed glial-specific genes, comprised of two-thirds of all differentially expressed genes (DEGs). We found microglial DEGs that were the most numerous in LPS group. On the contrary, differential alternative splicing (DAS) analysis revealed the most numerous DAS events in astrocytes. Besides, we observed distinct major isoform switching in the *Ptbp1* gene, with skipping of exon 8 in LPS group. *Ptbp1* usually considered a pluripotency sustaining agent in brain embryonic development, according to the previous studies. Analyzing the splicing tune-up upon LPS exposure, we came to a supposition that the short *Ptbp1* isoform de-represses immune-specific response by *Ptbp1* adjusted splicing architecture. Additionally, the *Ptbp3* (*NOD1*) immune-specific splicing factor has apparently been de-repressed by the *Ptbp1* short isoform in glial cells. Notably, both the *Ptbp1* and *Ptbp3* genes express primarily in microglial/endothelial brain cells. We also report immune-related genes, altering their major isoforms upon LPS exposure. The results revealed immune modulating role of alternative splicing in brain.

## 1. Introduction

An increasing interest in the study of neuroinflammation is explained by its association with different neurodegenerative diseases. Inflammatory markers were detected in the rat brain even 2 years after the insult that can induce the neuropathology of Alzheimer’s disease type [1]. In order to understand the molecular changes that contribute to the development of these diseases, lipopolysaccharide (LPS), the component of the outer membrane of Gram-negative bacteria, is widely used for induction of the central inflammatory process [2]. Injections of LPS also resulted in gene expression changes in brain structures, including the hippocampus [3], brain region that was implicated in the development of neurodegenerative-related psychopathology. Inflammatory- and apoptosis-related genes were revealed among these genes.

Alternative splicing manifests the flexible and rapidly evolving mechanism [4,5] of expanding the proteome diversity. The diversity is particularly vivid in the nervous system [5]. Neuroinflammation related CD44 alternative splicing was shown to affect the response of the hippocampus in the Alzheimer disease [6]. The synaptic proteins of *Nrxn1−3* were shown in response to neuroinflammation by altering their isoforms ratio [7].

Along with the mature neurons, there are stage specific splicing events featuring embryonic brain development. In particular, the previous studies have identified splicing factor *Ptbp1* as one of the major factors affecting the transition from neural stem cells (NSC) to neural progenitor cells (NPS) in the course of brain-specific embryogenesis [8,9,10].

Recently, the study reported *Ptbp1* mediating inflammatory secretome and tumorigenic processes by altering splicing landscape in senescence cells [11,12] and tumors [13], exhibiting *Ptbp1* as an immune repression factor. Recent study featured sophisticated interplay of *Ptbp1*, *Ptbp2*, *RbFox2*, and *SON* (SON DNA and RNA binding protein) promoting glioblastoma multiforme (GBM) genesis [14].

We addressed the impact of DAS genes in our study, assuming the major players of LPS response are astrocytes, microglia, and endothelial cells, along with consequent neuron adaptation. We analyzed the overall dynamics accounting for *Ptbp1* alteration and found *Ptbp1* exon 8 skipping may modulate immune competent response, primarily in microglial, astrocyte and endothelial cells in hippocampus.

## 2. Materials and Methods

### 2.1. Animals

Adult male Wistar rats (2.5 months of age) were used in the experiments. Animals were housed individually in polycarbonate cages (27.7 × 44 × 15 cm = w × l × h) with free access to food and water.

All animal use procedures were supervised and specifically approved by the ethic committee of the Institute of Cytology and Genetics, in accordance with the guidelines of the Ministry of Public Health of Russia (supplement to order N 267 of 19 June 2003) and European Council Directive (86/609/EEC). The middle cerebral artery occlusion and LPS administration into striatum included all measures to minimize rat suffering.

### 2.2. LPS Administration

The global agenda of our research is getting an insight on the involvement of stroke-induced inflammatory activation on the hippocampus by determining the genes directly affected by pro-inflammatory stimuli. Since middle cerebral artery occlusion (MCAO) causes the most severe neuronal damage in the ipsilateral striatum in rats [15] we have chosen the striatum for central LPS administration, thus complying to the published protocol for an acute rat model of local neuroinflammation in this brain structure [16].

LPS (30 µg in 4 µL of sterile saline) from Escherichia coli, serotype 055:B5 (Sigma-Aldrich Corp., St. Louis, MO, USA), or an appropriate volume of saline (SAL) were infused stereotactically into the right striatum under isoflurane anesthesia (4% isoflurane for induction, 2.5% for maintenance in O2 at a flow rate of 1 L/min). We used the following coordinates for drug infusions: AP = +0.5 mm, ML = +3 mm, and DV = −5.5/4.5 mm [16]. As was shown previously, this LPS treatment regimen also effectively provoked an acute neuroinflammation in the rat hippocampus [15].

### 2.3. Collecting Hippocampal Samples

Twenty-four hours after LPS administration, the rats were sacrificed by rapid decapitation. Brains were quickly extracted and ipsilateral hippocampi (*n* = 3 for each group of LPS, SAL) were rapidly isolated, and each was placed in an Eppendorf tube with 1 mL of buffer containing an RNase inhibitor (RNAlater) at room temperature. After that, the tube was transferred to ice, after the end of hippocampal collection, stored overnight at +4 °C, and then at −80 °C until the analysis of gene expression patterns.

### 2.4. RNA-Sequencing and Data Analysis

RNA-seq was performed in JSC Genoanalytica (Moscow, Russia; Available online: http://genoanalytica.ru, accessed on 15 November 2021). For this, total RNA was extracted from the ipsilateral hippocampus in the ischemic and LPS-infused rats with Trisol reagent, according to manufacture instruction. Quality was checked with the BioAnalyser and RNA 6000 Nano kit (Agilent, Santa Clara, CA, USA). PolyA RNA was purified with Dynabeads^®^ mRNA Purification kit (Ambion, Austin, TX, USA). Illumina library was made from polyA NEBNext^®^ Ultra™ II RNA Library Prep (NEB, Ipswich, MA, USA), according to manual. Sequencing was performed on HiSeq1500, with 50 bp read length. At least 30 million of reads were generated for each sample.

The raw reads from RNA-seq experiments were trimmed for quality (phred ≥ 20) and length (bp ≥ 32) using the Trimmomatic v. 3.2.2 [17]. Reads were mapped to the Rnor_6.0 genome with STAR aligner [18] and differentially expressed transcripts were inferred by Cuffdiff software v.2.1.1, accessed on 10 May 2021, http://cole-trapnell-lab.github.io/cufflinks/manual/ [19]. Genes with an adjusted p value (padj) less than 0.05 were classified as significantly differentially expressed genes (DEGs). A total of 34,600 transcripts were in the reference (annotation) set.

Differential alternative splicing (DAS) analysis has been performed using rMATs software, version 4.2.2., accessed on 10 May 2021 http://rnaseq-mats.sourceforge.net/download.html [20]. Only exon skipping (ES) events were considered.

### 2.5. Statistical Methods

GO enrichment analysis has been performed, with the string-db.org routine (string-db.org; accessed 15 December 2021). GO enrichment non-redundant grouping was performed with GOMCL software [21], accessed on 10 October 2021; https://github.com/Guannan-Wang/GOMCL).

Principal component analysis (PCA) has been performed by commercial XLSTAT software (accessed on 10 May 2021; https://xlstat.com). As a distance matrix, a Pearson pairwise correlation matrix of DEGs expression profiles was used. Agglomerative hierarchical clustering (AHC) was employed using XLSTAT software. The AHC parameter set was: (similarity: Pearson correlation coefficient; agglomeration method: unweighted pair-group average; center: no; reduce: no; truncation: automatic—inertia). Heatmap construction was performed with a self-organizing map (SOM) algorithm employing XLSTAT software.

## 3. Results

### 3.1. GO DEGs Analysis

Due to the brain invasion and consequent post-operational inflammation in both samples (SAL and LPS), we filtered the DEGs sample for the genes maintaining abs (log2 fold) >2. This resulted in final DEGs set of 87 entries (Appendix A).

Analysis of the gene ontology (GO) in the biological process yielded 284 entries (Appendix A). To obtain the core DEGs set, we applied a GO clustering algorithm GOMCL (Wang et al., 2020), which yielded just one GO supercluster consisting of 37 distinct DEGs, encompassing the 284-fold initial GO BP annotation list (Appendix A). This way we were able to see the non-redundant DEGs network, represented in Figure 1.

We employed PCA analysis (Figure 2a) outlining three clusters depicted in Figure 1 to proceed on details of each cluster. We found that *Hcrt*, attenuated in the LPS group, refers to hormone activity (Figure 2b), while two other clusters manifest inherent distinct coordinated responses to LPS exposure (Figure 2c,d). Elucidating glial-specific genes in the cluster using brain cells expression atlas, presented in [22] we were able to relate 24 genes as glial-specific ones (Figure 2e,f).

We found *Shox2* DEG, induced by an outlier value in the SAL1 sample, equaling 7.4 FPKM, while five other samples maintaining it as less that 0.07 (avg = 0.05; stddev = 0.03), so we discarded it.

Overall, we came to 36 DEGs as the core DEGs spanning the GO spectra. Microglial-specific genes were prevalent (22 DEGs) followed by endothelial genes (*Fos*, *Sele*; Figure 2e) and astrocytes display 2 genes (*Serping1*, *Fcnb*). Notably, *Fos* DEG expresses virtually in all cell types, most prominently in astrocytes [22] implying the assignment of this gene to epithelial cells may be due to *Sele* DEG co-variation. Other DEGs were not immediately assigned to glial cells; still, more than half of DEGs (24 entries) were assigned to be glial rather confidently.

### 3.2. DAS Genes Analysis

We identified 49 differentially, alternatively spliced genes with ES event alteration FDR <0.05 between SAL and LPS groups, presented in Appendix A. Only one DAS event per gene was observed. While no significantly enriched GO terms were featured for the set, we ascribed them manually.

We observed 3 ncRNA and 46 coding RNAs, one currently not annotated in public repositories (AABR07039316.1) and overlapping two small coding genes. Manual annotation elucidated several functional categories: lipid and lipolysis related (three DAS genes: *Lsr*, *Irf3*, and *Lrp8*); splicing factors and chromatin rearrangement (*Ptbp1*, *Rnps1*, and *Rbm34*), mitosis associated centromere proteins (*Cep126* and *Cep295*), coagulation associated factors (*F8*, *vWa5b2*), and glycoproteins/development related one (*Eogt;* Notch3-signaling), AABR07019088.1 (AS RNA of *Nduf3b*).

A total of 49 genes, featuring DAS ES events, maintain 9 median isoforms per gene (Appendix A), as annotated in the NCBI repository. We observed 3 median isoforms per gene for the same 49 DAS genes in our RNA-seq data, when including all ES events detected. The Pearson correlation of the RNA observed and expected (NCBI) numbers is quite significant (equaling *r* = 0.665, df = 45 (excluding three non-annotated RNA), *p* < 1 × 10^−5^) and implying we maintain good sensitivity with our AS detection algorithm.

Pursuing further analysis and keeping in mind the previous studies on neuroinflammastion, we suppose that: (a) the major role in LPS response attributed to glial cells; (b) non-random DAS ES events invoke the corresponding genes expression modulation (up or down).

#### 3.2.1. Cell-Specific Expression of DAS Genes

First, we assessed the distribution of 49 DAS gene (DAS FDR < 0.05) expression profiles across SAL and LPS samples (Figure 3). Upon analysis of Figure 3 we may state there is certain preference of DAS genes to SAL/LPS samples, based on their PCA of expression profiles, depicted in Figure 3. In particular, 17 glial DAS genes augmented their expression in the LPS group (Figure 3; bold typed genes, orange circles), while only 6 glial DAS genes attenuated it in LPS group (Figure 3; blue circles).

Based on [22], the brain cell-specific expression profiles we assessed showed 49 for cell specific expression (Figure 4). We identified the gene as a cell specific on when it maintains the highest expression among seven cell types. As expected, the most AS events took place in neuron specific genes (13–16 DAS genes). Still, we observed distinct glia specific genes in Figure 4 (colored gene names in the first column).

9 genes were not found in Atlas. In particular, *cep126*, *cep295* are absent, while 21 *cep* genes family was found highly expressed specifically in endothelial cells (order of magnitude higher than in any other cell type), but further were omitted from the analysis due to high non-specificity to LPS issue. No noncoding (ncRNA)/unannotated ones were found in Atlas, in particular: *Spaca*, AABR07019088.1, AABR07039316.1, AABR07069473.1. *Ogt* was used for annotation of *Eogt*, *Stk30 for Mok.* Both *Stk30* and *Ogt* are astrocyte specific genes. Thus, we maintained 42 genes as annotated for cell specificity.

#### 3.2.2. Sampled DAS Genes

Based on Figure 4 we defined 3 glia specific DAS groups comprising 17 genes. To further confirm their specificity within glial cells we built PCA plot for 3 glial cells (Figure 5). Most of the DAS genes were observed in astrocytes, according to Figure 5. Overall, we compiled 24 glial-specific DAS genes presented in Table 1.

### 3.3. Microglial Genes

We annotated *Ptbp1*, *Rhog*, *Atp13a2*, and *Afmid* according to Table 1.

#### 3.3.1. Rhog Gene

*Rhog* (Ras homolog gene family, member g (rho g)) coordinates response to ‘bacterial invasion of endothelial cells’ as is underlined by GO enrichment term (Figure 6).

We observed *Rhog* DAS ES event as 5′ UTR exon 4 skipping (Figure 6; right panel) implying it expression elevation in LPS group (Figure 3). Additionally, we observed distinct elevated expression of long isoform in LPS compared to SAL group, along with higher expression of short one (Figure 6; right panel). Exon 4 skipping is manifested as a major isoform in both SAL/LPS groups (Figure 6).

Figure 7 demonstrates non-linear coordinated elevation/downturn of *Atp13a2*, *Afmid* isoforms expression (Figure 3).

#### 3.3.2. *Ptbp1* Exon 8 Skipping Preference

Strikingly, we also observed significant exon 8 skipping in neurospecific splicing factor, *Ptbp1* (Figure 8), responsible for splicing alteration of several thousand genes [23]. Along with its expression elevation in LSP (Figure 9), its splicing skews towards to exon 8 skipping (Figure 5) implies putative immunomodulation in astrocytes/microglia/endothelial cells in LPS1 species, since *Ptbp1* short isoform weakens its polypyrimidine tract binding [23]. As exon 8 skip in *Ptbp1* alters target exon inclusion rate based on *Ptbp1* binding preference [23], this way altering exons inclusion rate (both up and down), we may speculate it serves as a switch to turn on immune response by altering exon inclusion/skipping events ratio in specific genes.

Based on annotation of 4 microglial cell specific genes we may state that DAS underscores specific major isoform enhancing. Three genes (*Ptbp1*, *Atp13a2*, and *Rhog*) have their expression rate increased along with altering isoforms ratio (Figure 3), while *Afmid* decreased its expression in LPS (Figure 3) featuring skipping of coding exon and thus being a subject to nonsense mediated decay (NMD) due to ORF disruption.

### 3.4. Endothelial Specific DAS Genes

Herein we annotate the endothelial cells specific DAS genes (Table 2): *Cast*, *Phactr4*, *Nebl*, *Lrp8*, and *Pkn2.* Figure 9, Figure 10, Figure 11, Figure 12 and Figure 13 underline the relevance of DAS genes to immune modulated features.

#### 3.4.1. *Cast* Gene

*Cast is* calcium-dependent cysteine protease. Based on annotation (Figure 9), we may state that *cast* is involved in ‘degradation of extracellular matrix’ and may be implicated in immune response processes.

#### 3.4.2. *Phactr4* Gene

*Phactr4* gene was shown to be involved in mediation of transient response potential regulation upon inflammatory response (Figure 10a) featuring short isoform expression elevation in LPS sample (Figure 3 and Figure 10b).

#### 3.4.3. *Pkn2* Gene

*Pkn2* (Serine/threonine-protein kinase N2) is involved in endothelial cell migration (GO:0010594), also featuring intracellular signal transduction in response to stress (Figure 11). DAS ES event in this gene features its activation by elevation major long isoform in LPS group leading to its enhanced expression (Figure 3).

#### 3.4.4. *Ndufb3* vs. *AABR07019088 Genes*

*Ndufb3* is a member of MWFE subunit of the mitochondrial NADH-ubiquinone oxidoreductase (complex I) is a small, essential membrane protein of 70 amino acids, which is made in the cytosol, imported into mitochondria, and assembled without further proteolytic processing. *AABR07019088* is antisense RNA of *Ndufb3* (Figure 12a). Both genes elevated their long isoforms expression in LPS group (Figure 3 and Figure 12b), *Ndufb3* isoform featuring alternative translation start site (TSS; Figure 12a).

#### 3.4.5. *Pan3* and *Nebl* Genes

*Pan3* (Poly(A) Specific Ribonuclease Subunit) and *Nebl* (Nebulette; Actin-Binding Z-Disk Protein) are housekeeping genes. Long isoforms elevation (Figure 13) may witness enhanced metabolism in LPS group (Figure 3).

### 3.5. Astrocyte Specific DAS Genes

Astrocyte specific DAS genes: *Ank2*, *Med12l*, *Tead1*, *Milt10*, *Fnbp1*, *Zfp1*, *Wrd78*, *Eogt*, *Dctd*, *Phc3*, and *Mok* (Table 2) manifest the largest group, with two genes (*Ank2*, *Eogt*) maintaining the highest DAS significance.

We observed the two most significant DAS ES events (Appendix A) in *Ank2* (DAS FDR < 0.006) and *Eogt* (DAS FDR <0.00014) genes altering its major isoforms upon LPS exposure (Figure 14). Both these genes also express in neurons (Figure 4). Notably, while *Ank2* increased its expression in LPS, *Eogt* was attenuated (Figure 3), due to 5′ UTR related exon skipping implying glycosylation process downturn.

The rest of DAS genes referring mainly to transcription process and apparently featuring metabolism intensity increase in LPS (Figure 3), all manifesting long isoforms in LPS group, are presented in Figure 15.

#### *Fnbp1* Gene

*Fnbp1* is Formin binding protein 1. It is required to coordinate membrane tubulation with reorganization of the actin cytoskeleton during the late stage of clathrin-mediated endocytosis. (DAS FDR < 0.027). We observed its major isoform elevation, accompanied with attenuation of alternative skipped isoform (Figure 16).

### 3.6. Splicing Factors (SF) in Brain

We assessed 15 key SFs expression in the brain using Barres Lab atlas for mature brain cell specific expression [22]. As can be seen from Figure 17, the majority of brain-specific SFs are neuron specific ones, as was reported earlier, in a range of publications starting from 2005 [24,25]. Still, there is a range of SFs, in particular, *Ptbp1*, expressing preferentially in microglia and endothelial cells (Figure 13). As we observed, *Ptbp1* exon 8 skipping as highly specific for LPS exposed species (Figure 4), we speculated it might affect immunomodulation in response to LPS.

From Figure 18 we may state that glia specific SFs (Figure 18a) are elevated in LPS sample (Figure 18b), implying its involvement in LPS response.

## 4. Discussion

We conducted RNA-seq analysis of rat hippocampal tissues upon LPS exposure compared to saline administered ones. Analysis revealed 609 genes with significant difference in expression rate at *q*-value < 0.05, which were further filtered with log2 fold > 2 criterion, due to surgical invasion procedure in both SAL/LPS groups and consequent background inflammation in both groups (Appendix A). That left 93 DEGs that were subject to gene ontology annotation, yielding 284 GO biological process terms. Using GOMCL toolkit [21], we elucidated single DEGs cluster based on ontology and consequent non-redundant 36 DEGs, encompassing all GO terms present in initial sample.

We present the key relevant events in Table 2, while full GO annotation can be found in Appendix A. Based on Table 3 we may state we witness quite clear LPS effect within our sample compared to SAL control group. The 3 major clusters of coordinated genes.

Based on DEG analysis, (Figure 1 and Figure 2) we state that LPS response is quite robust (612 DEGs overall) and single directed (majority of DEGs are elevated in LPS group; Figure 2). The 36 core DEGs co-varied clusters are depicted in Figure 2. We should stress that the majority of DEGs are highly expressed glial genes. The cell specific analysis reveals the majority of DEGs are microglial specific ones following endothelial and astrocyte cells (Figure 2e,f).

Given AS in brain primarily refers to neurons primarily expanding their synaptic plasticity [26] and has been widely addressed therein; we decided on assessing AS-mediated LPS response specifically in glial cells for gaining the insight on it. Analysis of alternative splicing reveals a group of 50 structural and signaling genes being differentially alternatively spliced (see Appendix A). No particular abundance in GO categories were inferred for the DAS gene set. Still, while expanding the gene neighborhood of DAS genes there were relevant DAS genes networks affecting rho-gtpase activity, platelet-related genes (*F8*, *wVag1*; [27]) and a range of others (Figure 6, Figure 7, Figure 8, Figure 9, Figure 10, Figure 11, Figure 12, Figure 13, Figure 14, Figure 15 and Figure 16).

With that, we should outline the basic challenges of splicing analysis we have observed. Due to relatively low read coverage in our study (about 30 mln reads per transcriptome), we might miss some DAS genes, along with maintaining DAS low confidence (sensitivity) in a range of cases. Recently reported tandem splice sites (TASS) persistence reported in RNA-seq data [28] were also observed in our data quite abundantly (about 1.15 thousand, across 10,000 ES events total average per sample) given only 20% proved being non-spurious/sensible splice sites [28], also distorted the overall picture. We applied the filter of minimal exon junction counts to be more than five, totaling both groups, compared.

Additionally, based on our preliminary assessments with protein database, at least half of the ES events maintain open reading frame (ORF) disruption and invoking nonsense mediated decay (NMD), thus being purely the means of prompt regulation of homeostatic balance of particular genes expression [29].

Another fundamental problem is that, in contrast to expression temporal course, the splicing dynamics manifests a much more velocity, implying there are large pools of unspliced transcripts in the pool [30] waiting for the splicing ‘decision’. Splicing alteration may be performed in a matter of milliseconds and be restored back immediately, especially for many auto-/cross-regulated splicing factors and chromatin rearrangement machinery by utilizing NMD routine while altering (downing) gene expression would take relatively longer time.

Functional interpreting ES events is also a keystone. We approached this subject at least by validating coding/non-coding transcript variants resulted in the course of ES events. We revealed that only 8 DAS genes (*Ptbp1*, *Fnbp1*, *Med12l*, *MegF11*, *Cast*, *Phc3*, *Cep295*, and *Ank2*) maintain both coding isoforms, 19 DAS genes maintain long coding isoform only, no genes with skipped isoform coding only, and 23 remaining DAS genes maintain ES events in 3′/5′ UTRs. Elaborating on coding ES renders protein interaction analysis and, while not being already annotated experimentally, is hard to exemplify. Still, new means of protein isoforms annotation are approaching with an advent of AlfaFold resource [31].

*Ptbp1* is one of the key neurogenesis factors, also used in reprogramming neural cells [8,32,33,34] We elucidated it is primarily expresses in glial cells (Figure 4), while highly repressed in neurons. Intriguingly, recent study demonstrated morphing astrocytes into dopaminergic neurons when depressing *Ptbp1* [34]. They did it with virus vectors while natural antagonist of *Ptbp1* in embryogenesis is *mir-124* [8]. With that, the highest expression of *Ptbp1* is observed in microglia and endothelial cells.

As we found essential splicing factor *Ptbp1* being significant DAS gene (FDR < 0.03) in our set, we decided exploring brain-specific splicing factors genes group, since these affect splicing the most and, while being altered and impact several thousand of relevant exons reported elsewhere [5,8,23]. Upon the analysis, we report glial SFs manifest augmented expression in LPS group (Figure 13) emphasizing their relevance.

The mammalian specific *Ptbp1* exon 8 (9 in human) skipping (27aa) manifests one of current enigmas not fully apprehended. It reported to be involved in embryonic stage tune-up after NSC -> NPC stage splicing tune up [23]. It is reported therein “exon 9 possesses splicing regulatory activity that is partially separable from the repressive activity conferred by RRM2 and that skipping of exon 9 reduces the negative and positive regulatory activities of PTBP1 without substantially affecting RNA binding activity”, implying significant effect of the ES9 event.

Notably, according to GTEX v.7 data (gtexportal.org), *Ptbp1* ES9 short isoform (ENST00000349038; NM_031991) is a major one with average expression rate of 31 transcripts per million (TMP) across 53 tissues, while exon 9-in most abundantly expressed variant (ENST00000350092; 5′ UTR truncated isoform) manifests 16 TPM average, with other 16 ones manifesting 3.5 TPM and less, still comprising overall around 40% of *Ptbp1* total expression rate per tissue. There is only one coding ES9 isoform in human *Ptbp1* isoforms spectra, except for extremely rare transcript ENST00000627714.2, which simultaneously lacks all RNA recognition motifs (RRMs) coding exons and not detected in GTEX data. Two top expressed *Ptbp1* isoforms co-vary with Pearson *r =* 0.81 (df = 52; *p* < 1 × 10^−7^).

Based on analysis of *Ptbp1* expression organism wide with GTEX human resource, we elucidated that it’s a redundantly expressed gene observed in virtually all cells except for mature neurons, and immune competent organs including heart, whole blood, pancreas, liver (Table 3a). Oppositely, cancer-related tissues/cells manifest high *Ptbp1* expression rates, as reported in studies [11,35]. *Ptbp1* deletion enhances MHC II expression in dendritic cells [13], therefore attenuating its binding affinity by ES8 may do the work. Based on glial cells analysis, we may state that *Ptbp1* is significantly expressed in epithelial cells, as well (Figure 4).

Given *Ptbp1* is primarily a glia specific gene among differentiated brain cells, we found its overall expression elevated specifically in LPS administered group, though not significantly (Figure 3. *p*-value < 0.33), at the same time featuring its major isoform, significantly altering toward ES8 one (Figure 8), implying some stress response activity. Expanded *Ptbp1* isoforms spectra (17 entries) allows for proposing *Ptbp1* isoform expression ratios, which may impact the immunogenic potential of glial cells, judging by the dynamics it manifests upon LPS response in the hippocampus brain region.

## 5. Conclusions

While many of DAS events relate to the modulation of expression rates (increasing/decreasing), we may underline the enhanced accuracy of splicing routine in the course of gene expression elevation (often non-significant one), since short isoform (usually non-coding one) preference often manifest attenuation in a vast range of cases. We report that, while DAS genes and DEGs do not overlap, as a rule (e.g., [36]), it was observed that NMD employed coding gene expression abrogation correlates with its expression rate downturn, implying certain feedback routines at the transcription layer [37,38]. To this end, we observe certain glial DAS genes change its expression rate in nonrandom manner (Figure 3).

While we cannot immediately, functionally interpret the AS ES events observed, besides overall expression rate alteration, we correspond that there are plenty of statistically significant AS ES alterations in glial genes, in particular. Underlining their abundance, further research may shed light on the functional meaning of DAS ES events, besides NMD-related ones. We stress the DAS of *Ptbp1* as one of the potent events possibly leading to massive alteration of splicing landscape upon LPS administration.

## Figures and Tables

**Figure 1 biomolecules-12-00277-f001:**
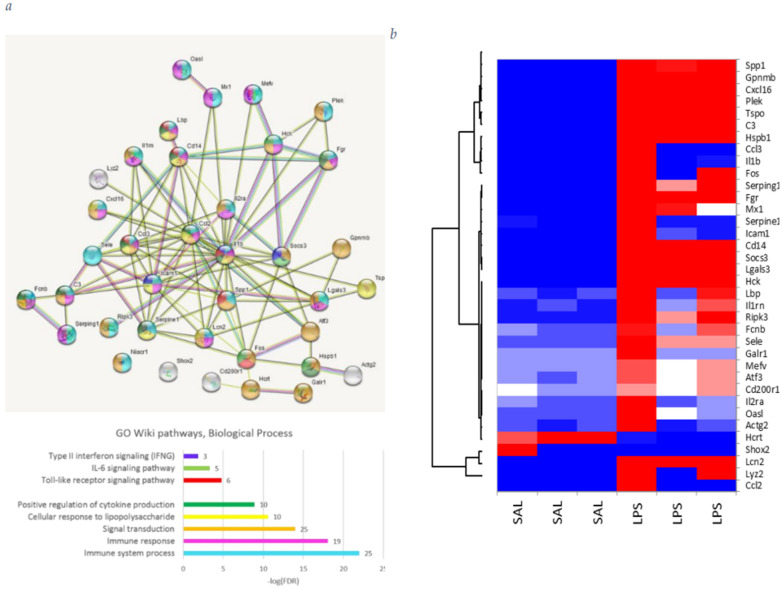
(**a**) GO annotation, 37 core DEGs in SAL vs. LPS comparison. Selected are several relevant GO biological process and wiki pathways from Appendix A. Colors on the plot correspond to those on the figure. Number of DEGs per GO term attached as bar labels. (**b**) Heatmap of 37 DEGs, clustering underlines three clusters: small one (*Hcrt* and *Shox2*), attenuated in LPS group, and two others (specific to the LPS group).

**Figure 2 biomolecules-12-00277-f002:**
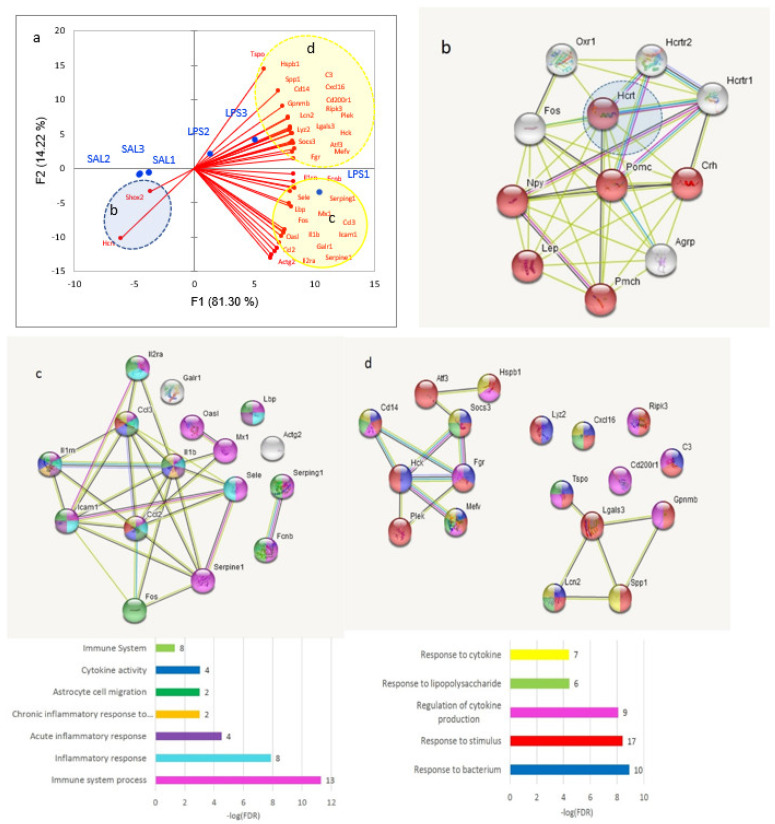
(**a**) 3 clusters (b, c, d) of 37 DEGs encircled by ovals; (**b**) SAL—related *Hcrt* environment unveils its relation to GO:0005179 (hormone activity; 6 genes of 108; FDR < 1.77 × 10^−8^), depressed in LPS; (**c**) coordinated mixed glial genes featuring microglial (*Serpine icam1*, *il1rp*, *il1b*, *ccl2*, *ccl3*, and *Fos*), endothelial (*Sele* and *Fos*), and astrocyte (*Serping*, *Fonb*, and *Fos*)) cells; 10 DEGs; (**d**) microglial coordinated cluster (16 DEGs); (**e**) PCA plot of (c) cluster DEGs across cell lines expression profiles [22]; (**f**) PCA plot of (d) cluster DEGs across cell line expression profiles. ‘Od’ stands for Olygodendrocyte.

**Figure 3 biomolecules-12-00277-f003:**
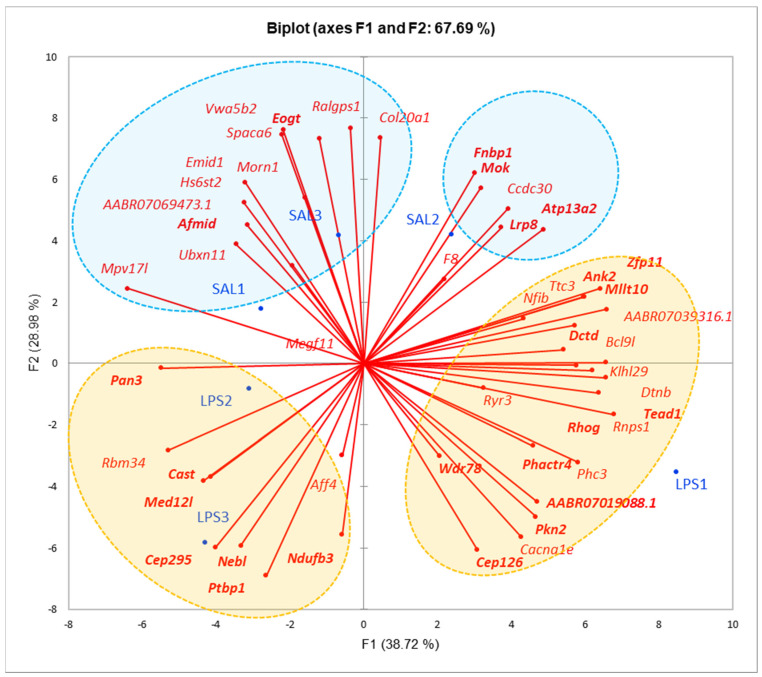
PCA plot, based on expression profiles of 49 DAS genes (FDR <0.05), across 6 samples, manifest certain preference to the samples. Orange shaded are DAS genes with enhanced expression in LPS, blue shaded are DAS genes attenuated their expression in LPS. Bold typed are glia-related genes (see text below).

**Figure 4 biomolecules-12-00277-f004:**
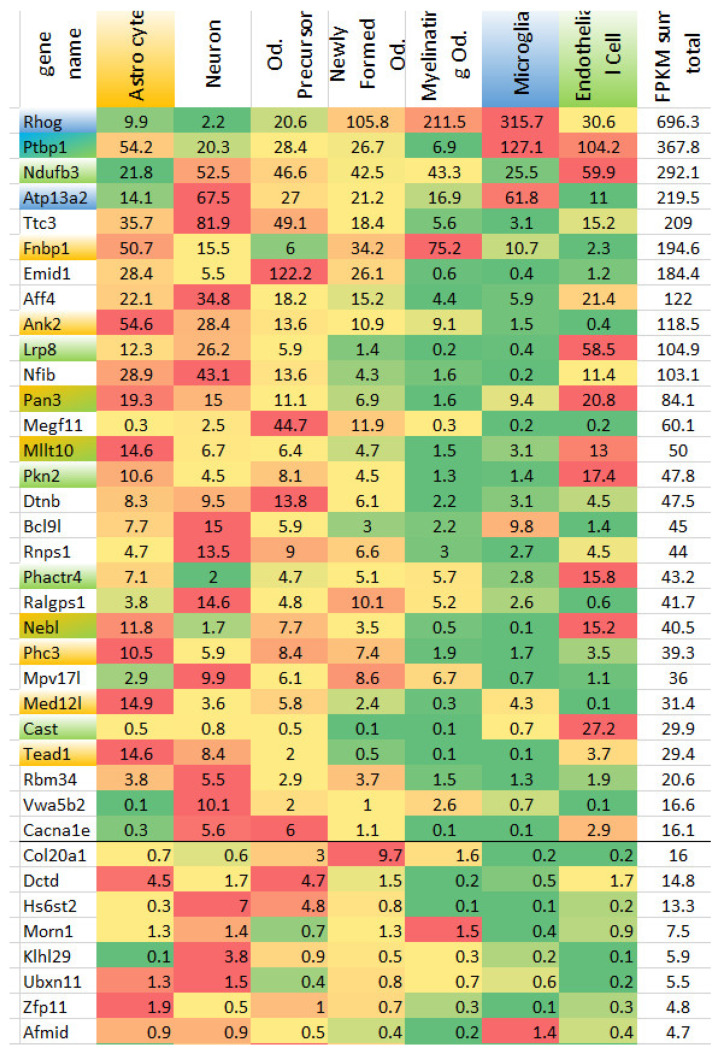
40 DAS genes (Figure 3), annotated according to Brain Cells Atlas (Zhang et al., 2014), elucidating glial-specific genes. Blue color of gene name (first column) corresponds to microglia; green—endothelial cell; orange—astrocyte. Double colors signify both cells maintain expression of the corresponding gene. Uncolored gene names correspond to other (non-glial) cell types. “Od.” abbreviation stands for “Oligodendrocyte”. Genes sorted descendent for overall expression rate (last column).

**Figure 5 biomolecules-12-00277-f005:**
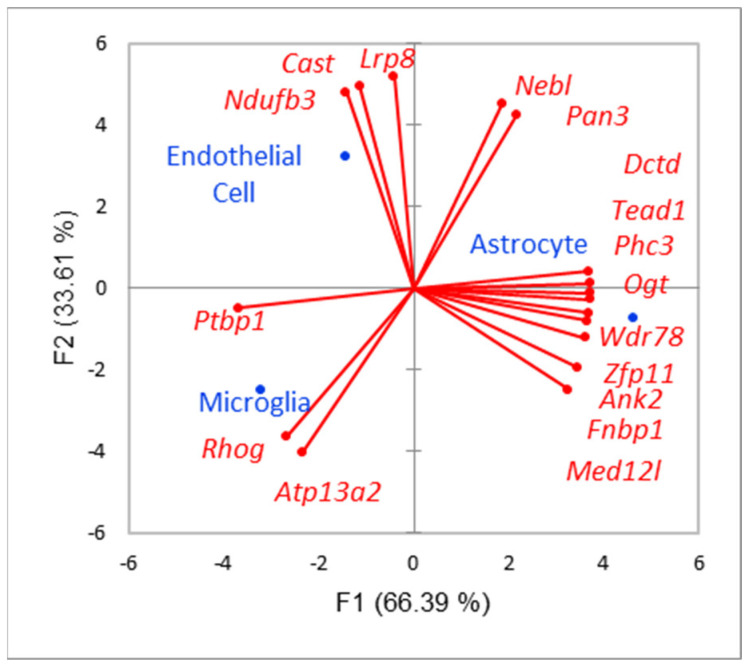
Selected 17 glia specific DAS genes.

**Figure 6 biomolecules-12-00277-f006:**
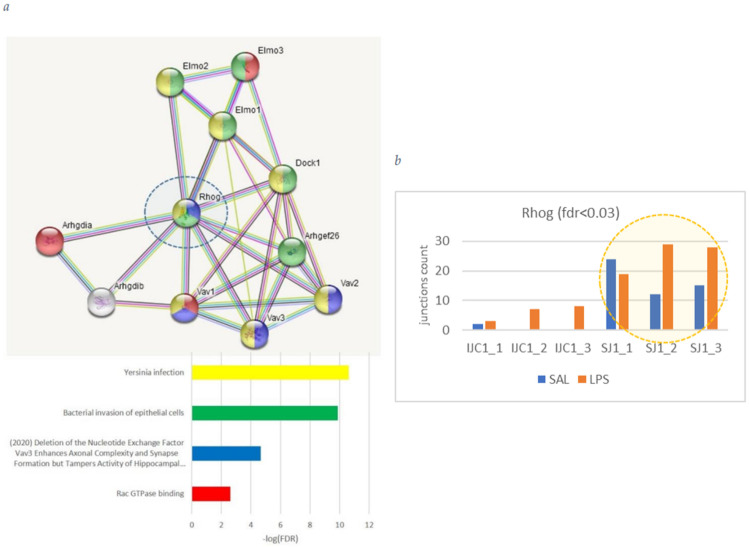
(**a**) *Rhog* genetic environment recovered by string-db.org facility. We observe it involvement in ‘Bacterial invasion of epithelial cells’ GO term, implying distinct LPS response. (**b**) Both *Rhog* 5′ UTR isoforms elevated (DAS FDR < 0.03); denotation: IJC (‘include junction counts’; Shen et al. 2014) stands for read counts overlapping exon inclusion instances for three replicas per group (ijc1, ijc2, ijc3); SJC (“skipped junction counts”) stands for read counts overlapping exon skipping junctions for each of three species per group.

**Figure 7 biomolecules-12-00277-f007:**
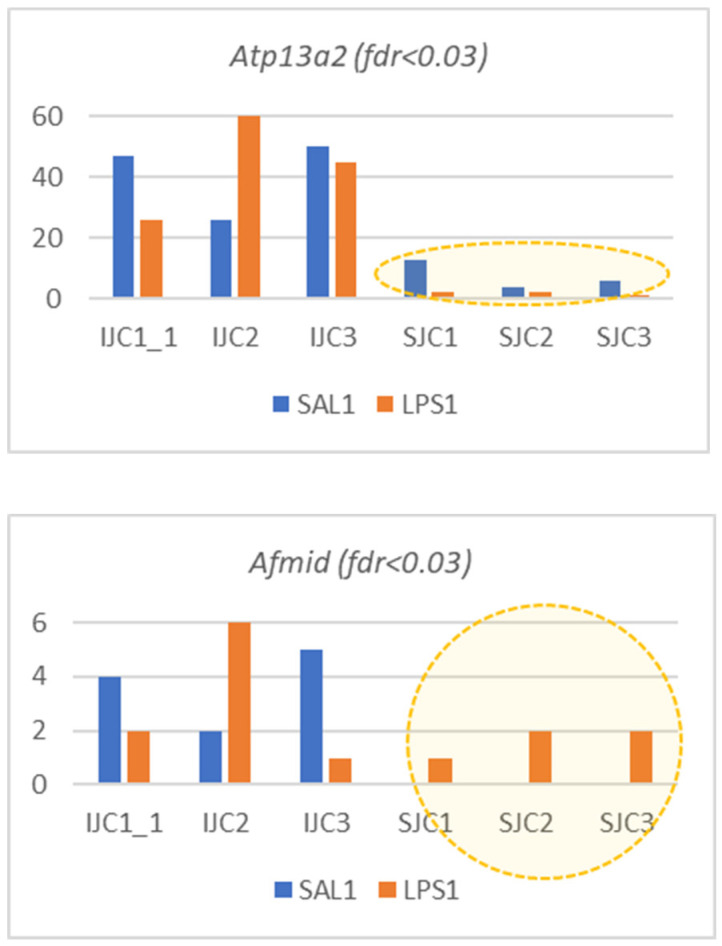
DAS ES events of microglial genes given *Atp13a2* increased expression in LPS, while *Afmid* decreased its expression (Figure 3). Short isoform is noncoding in both cases.

**Figure 8 biomolecules-12-00277-f008:**
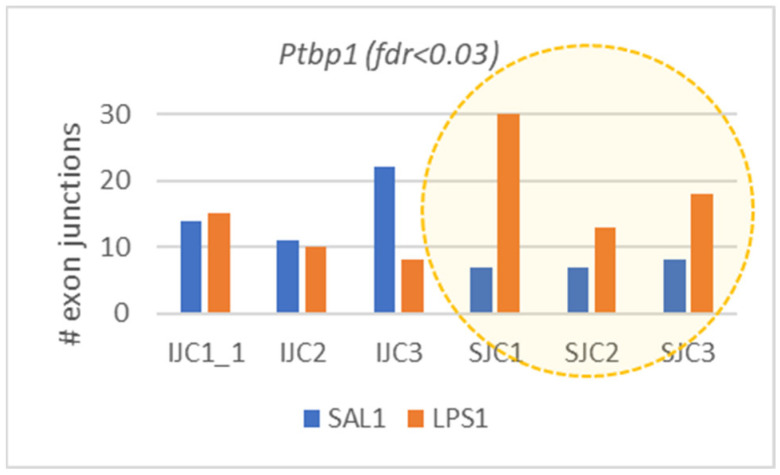
Alteration of major isoform from long to exon eight skipped one in LPS group, regarding to in-frame exon 8 (DAS FDR < 0.03).

**Figure 9 biomolecules-12-00277-f009:**
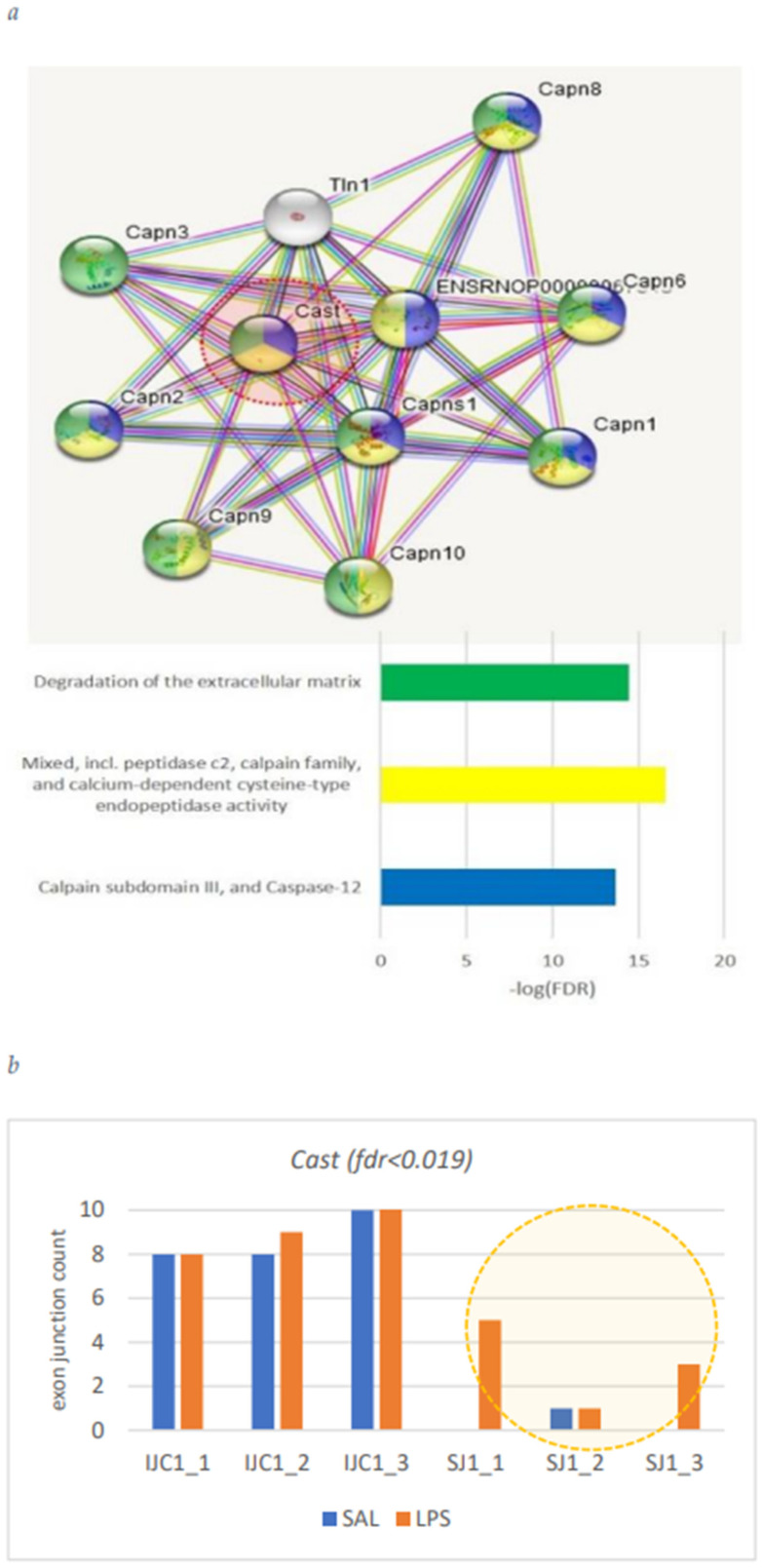
(**a**) Neighborhood environment of *Cast* and GO annotation. (**b**) Observed expression elevation of skipped isoform in LPS group.

**Figure 10 biomolecules-12-00277-f010:**
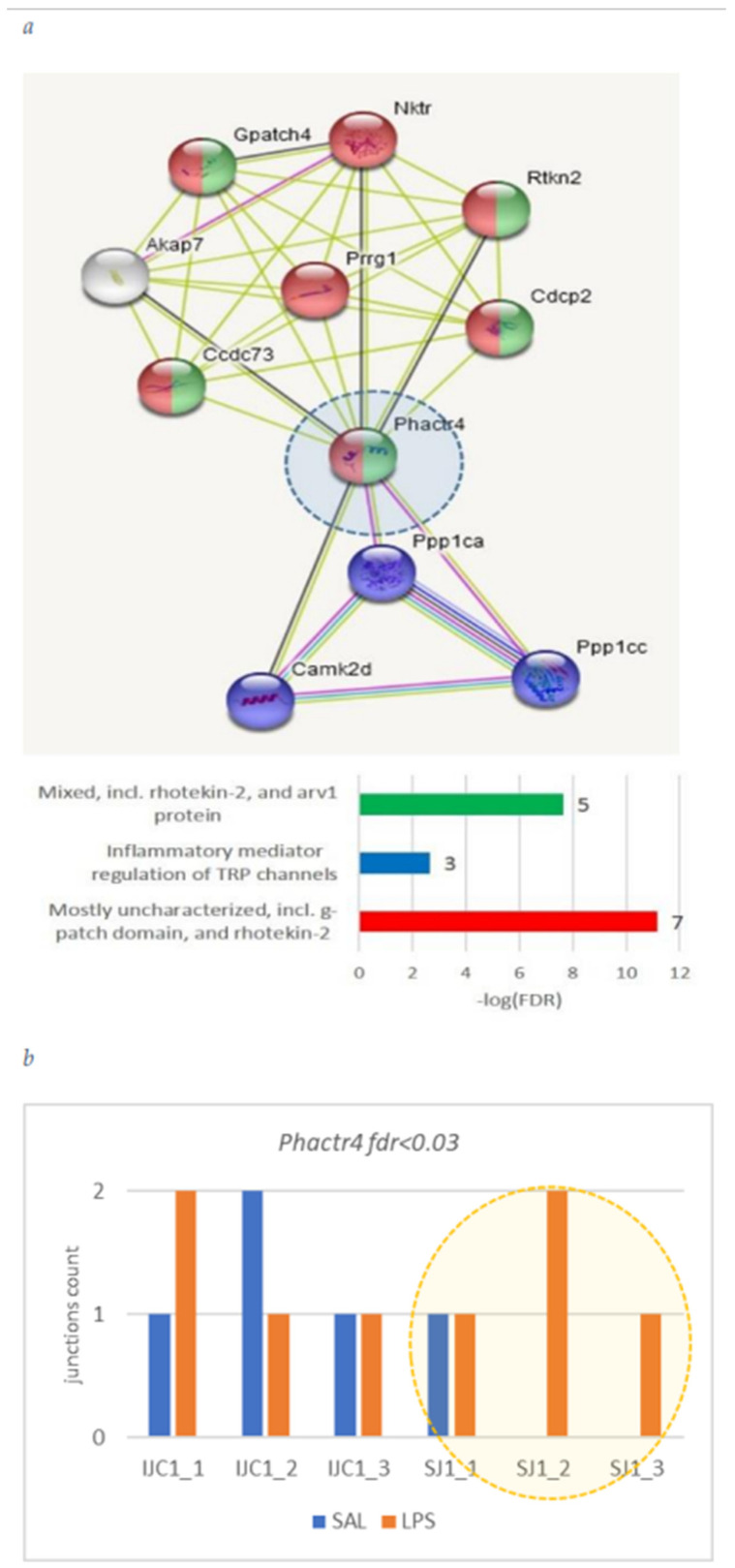
(**a**) String-db gene environment implies *Phactr4* mediating the inflammatory regulation of TRP (transient response potential) channels; (**b**) Short isoform is distinctly elevated in LPS group (Figure 3).

**Figure 11 biomolecules-12-00277-f011:**
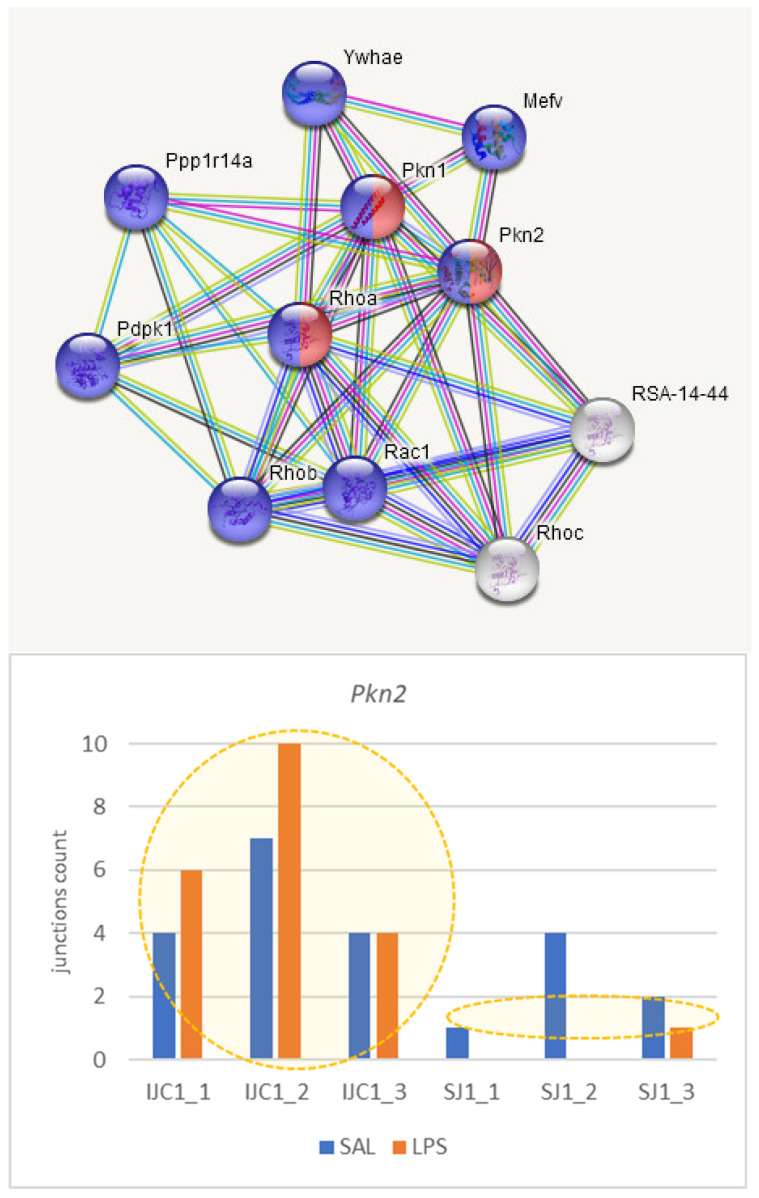
*Pkn2* is involved in ‘epithelial cells migration’ cascade (three genes, red; GO:0010631; enrichment FDR < 0.0017) and is a part of ‘Response to stimulus’ network (nine genes, blue; GO:0050896; enrichment FDR < 0.0037). DAS FDR < 0.03, AS enhances inclusion of coding exon in LPS sample.

**Figure 12 biomolecules-12-00277-f012:**
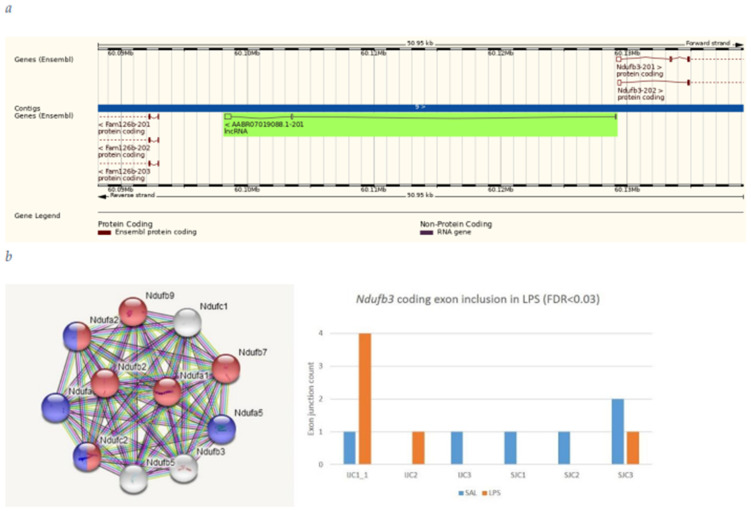
(**a**) Noncoding RNA *AABR07019088.1* manifests exon 2 elevated insertion (DAS FDR < 0.04; Appendix A) in *Nduf3b* antisense location along with *Ndufb3* one in LPS group; (**b**) Oxidoreductase MWFE subunit mitochondrial related gene *Ndufb3* switched TSS by coding exon inclusion (see (a)) in LPS group enhancing functional expression. Color coding: Red: GO: CL:22493; description: ubiquinone, and NADH-ubiquinone oxidoreductase MWFE subunit: obs vs. exp: 6 of 8; FDR: 2.53 × 10^−12^. Blue: GO: KW-0679, Respiratory chain; obs/exp: 4 of 40; FDR < 1.83 × 10^−6^.

**Figure 13 biomolecules-12-00277-f013:**
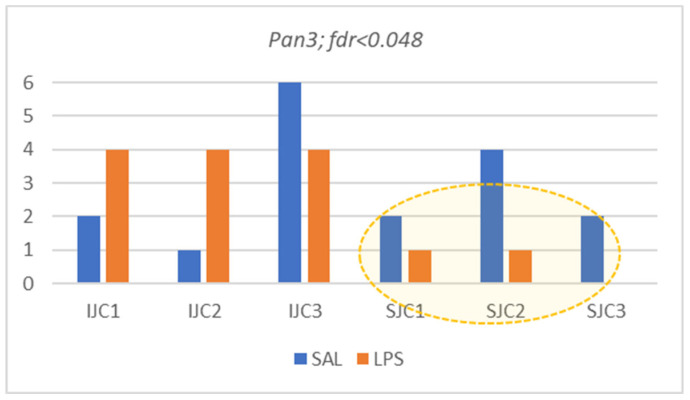
Both genes manifest elevated expression of long isoform in LPS (Figure 3) and attenuated expression of skipped (noncoding) ones.

**Figure 14 biomolecules-12-00277-f014:**
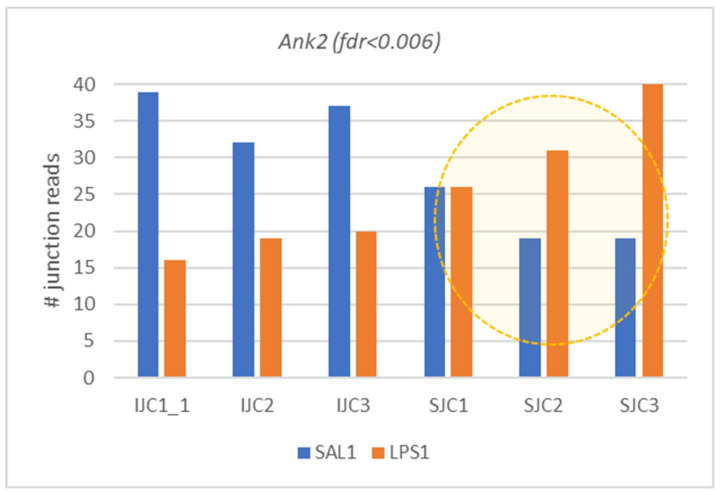
Switching major isoforms in two highly significant DAS genes. Ankyrine 2 (*Ank2*) gene increases short isoform expression upon LPS exposure, while *Eogt* Glycosyltransferase gene expressed, mostly in astrocytes, is switched off by DAS altering 5′UTR region.

**Figure 15 biomolecules-12-00277-f015:**
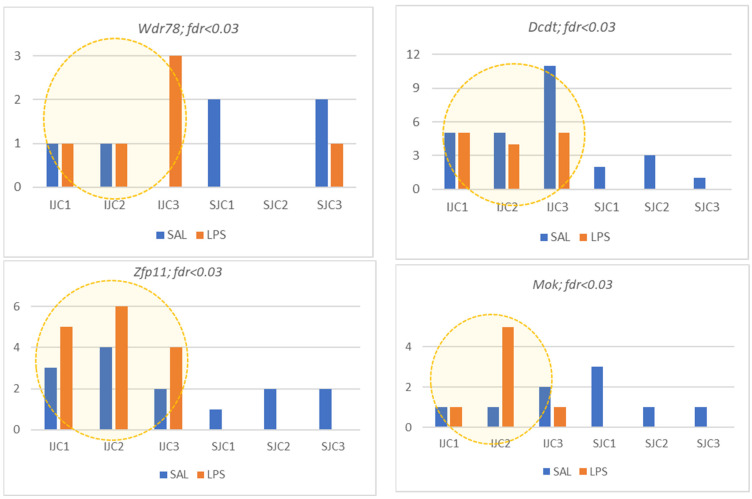
Panel of seven astrocyte specific low expressed DAS ES events. (*Wdr78*, Dynein Axonemal Intermediate Chain 4; *Dctd*, Deoxycytidylate Deaminase; *Zfp11*, transcription factor; *Mok*, MOK protein kinase; *Med12L*, Mediator Complex Subunit 12L; *Mllt10*, Histone Lysine Methyltransferase DOT1L Cofactor; *Phc3*, Polyhomeotic Homolog 3).

**Figure 16 biomolecules-12-00277-f016:**
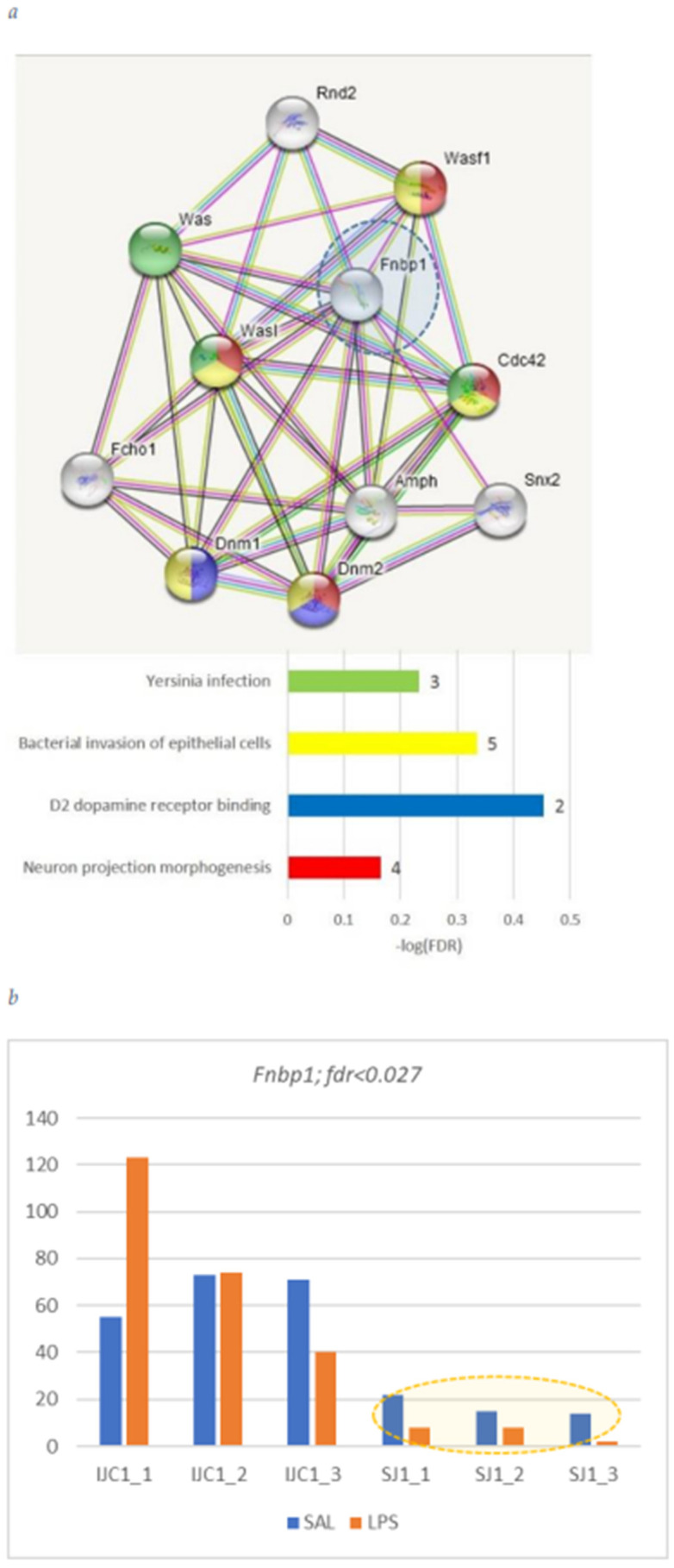
(**a**) *Fnbp1* mediates ‘Bacterial invasion of epithelial cells’ response as well via specific networks. (**b**) We see attenuation of short isoform in LPS group (DAS FDR <0.027).

**Figure 17 biomolecules-12-00277-f017:**
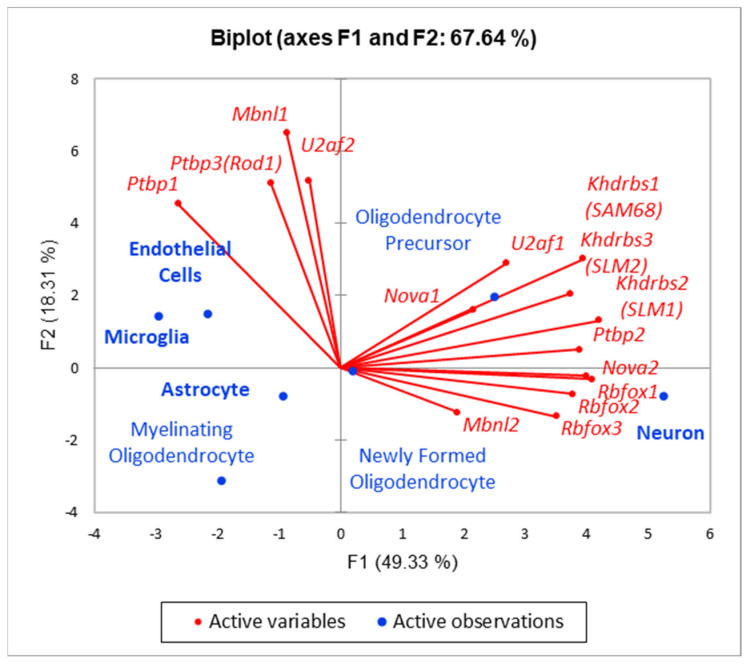
SFs expression preference in seven brain cell types [22]. Besides mostly neuron specific SFs (*Nova*, *Ptbp2*, *Rbfox*, *SLM1*, and *2*) we may see four SFs preferential ones in glial cells (*Ptbp1*, *Rod1*, *Mbnl1*, and *U2af2*).

**Figure 18 biomolecules-12-00277-f018:**
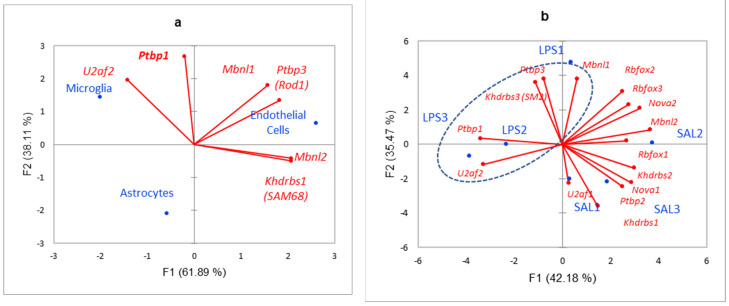
(**a**) PCA plot elaborating on glia specific SFs [22] underlined in Figure 17 across three glial cell types point to microglial and endothelial cells SFs activity, while astrocytes essentially lack SFs considered; (**b**) PCA plot of SFs expression profiling across 6 species underlining expression elevation of glial SFs specifically in LPS1 (encircled by blue oval).

**Table 1 biomolecules-12-00277-t001:** The list of 24 DAS genes preference in glial cells. Top significant DAS genes are bold typed.

Endothelial Cells	Microglia	Astrocytes
*Ptbp1*	*Ptbp1*	** *Ank2* **
*Cast*	*Rhog*	*Med12l*
*Phactr4*	*Atp13a2*	*Tead1*
*Nebl*	*Afmid*	*Pan3*
*Lrp8*		*Milt10*
*Pkn2*		*Ptn2*
*Ndufb3*		*Fnbp1*
		*Zfp1*
		** *Wrd78* **
		** *Eogt* **
		*Dctd*
		*Phc3*
		*Mok*

**Table 2 biomolecules-12-00277-t002:** Key SAL_LPS DEG networks inferred by GO annotation.

GO_id	Definition	Obs/Exp	FDR
GO: 0002366	Immune system process	25 of 945	9.70 × 10^−23^
GO: 0006955	Immune response	19 of 506	8.32 × 10^−19^
GO: 0007165	Signal transduction	25 of 2142	1.03 × 10^−14^
GO: 0071222	Cellular response to lipopolysaccharide	10 of 153	2.39 × 10^−11^
GO: 0001819	Positive regulation of cytokine production	10 of 239	1.18 × 10^−09^
GO: 0001817	Regulation of cytokine production	15 of 390	1.84 × 10^−08^
GO: 0032496	Response to lipopolysaccharide	14 of 306	1.26 × 10^−08^

**Table 3 biomolecules-12-00277-t003:** Minimal (**a**) and maximal (**b**) expression rate (TPM) of *Ptbp1* major isoform (ES9) in 10 tissues (source: GTEX v.7). Bracketed is top/second isoforms ratio. Median *Ptbp1* expression value across 53 tissues is 29.5 TPM.

(a) Pancreas	Liver	Whole Blood	Heart—Left Ventricle	Brain—Frontal Cortex (BA9)
9.4 (1.01)	8.42 (1.7)	7.39 (3.1)	6.28 (1.6)	6.72 (4.5)
(b)				
**Adipose—Subcutaneous**	**Lung**	**Cells—Transformed Fibroblasts**	**Cells—EBV-Transformed Lymphocytes**	**Cervix—Ectocervix**
82.4 (3.1)	85.5 (2.6)	80.5 (2.4)	101.4 (4.5)	57.3 (1.9)

## Data Availability

Data supporting reported results can be found in European Nucleotide Archive (ENA) with id: PRJEB50635.

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
