# Peer review of "LPS Administration Impacts Glial Immune Programs by Alternative Splicing"

_biomolecules, 2022, doi:10.3390/biom12020277_

Round 1

Reviewer 1 Report

Novel provided information may be useful in the research focused on microglia in infection/inflammation.

<Major points>

Authors used direct injection of LPS into brain, which is good method than peripheral administration of LPS. Since in this system LPS directly stimulation brain components such as astrocyte, endothelial cells, communication among several components may affect brain immune system. How LPS affects these components in hippocampus should be discussed. In addition, If you have in vitro using cell line, these data support your claim.

<Minor points>

Some typographical errors are found in this manuscript (page 11; Along withs its expression elevation in LSP (Fig. 9) and so on). Please amend these errors.

Author Response

Novel provided information may be useful in the research focused on microglia in infection/inflammation.

>Thank you

<Major points>

Authors used direct injection of LPS into brain, which is good method than peripheral administration of LPS. Since in this system LPS directly stimulation brain components such as astrocyte, endothelial cells, communication among several components may affect brain immune system. How LPS affects these components in hippocampus should be discussed. In addition, If you have in vitro using cell line, these data support your claim.

>We have significantly rewritten introduction, thank you. We have stressed the aim and scope of the research. We also provided some facts on the coding potential of DAS events in discussion shortly.

<Minor points>

Some typographical errors are found in this manuscript (page 11; Along with its expression elevation in LSP (Fig. 9) and so on). Please amend these errors.

>We did the spellcheck across the manuscript, thank you.

Reviewer 2 Report

  1. Introduction section: Please include more detailed description of neuroinflammatory regulations of LPS in the hippocampus.  
  2. Materials and Methods section: (1) Please provide the rationale why the authors inject LPS into striatum, not directly into hippocampus. (2) include coordinates for striatum (AP, ML, DV from bregma) (3) include manufactural  info of LPS (4) include   
  3. Results: authors should provide validation data of RNA sequencing via qPCR  
  4.  

Author Response

We are quite grateful for the valuable reviewer's comments and tried to respond to them as follows below.

  • Introduction section: Please include more detailed description of neuroinflammatory regulations of LPS in the hippocampus.  

>We have significantly rewritten introduction, thank you. Now we reference our previous studies providing more elaboration on the problem statement.

  • Materials and Methods section:
  • Please provide the rationale why the authors inject LPS into striatum, not directly into hippocampus.

Thank you for mentioning. The global agenda of our research was to get insight into the involvement of stroke-induced inflammatory activation in remote effects of stroke on the hippocampus by determining genes directly affected by pro-inflammatory stimuli. Since Middle cerebral artery occlusion (MCAO) causes most severe neuronal damage in the ipsilateral striatum in rats (Matsuda et al., 2009; PMID: 19238518), we have chosen striatum for central LPS administration according to the published protocol for acute rat model of local neuroinflammation in this brain structure (Ora et al., 2015; PMID: 26220690). Within this frame/protocol we pursue the impact of LPS induced by LPS to neuroinflammation on memory and hippocampus, which instantiates neuroinflammation response following striatum damage. Thus, we used the standard protocol of striatum mediated neuroinflammation evocation in complying with ischemic stroke modeling.

                        We added the short chapter on the issue at the beginning of LPS administration chapter for transparency, thank you.

Also we added: “As was shown previously, this LPS treatment regimen effectively provoked an acute neuroinflammation also in the rat hippocampus (Shishkina et al., 2021).”

(2) include coordinates for striatum (AP, ML, DV from bregma)

>Thank you, we added the following coordinates:

We used the following coordinates for drug infusions: AP= + 0.5 mm, ML= + 3 mm, DV= -5.5/4.5 mm (Ory et al., 2015).

(3) include manufactural  info of LPS (4) include   

>Thank you. We added:  “LPS (30 µg in 4 µl of sterile saline) from Escherichia coli, serotype 055:B5 (Sig-ma-Aldrich Corp., St Louis, MO, USA) or an appropriate volume of saline (SAL) were in-fused stereotactically into the right striatum under isoflurane anesthesia (4% isoflurane for induction, 2.5% for maintenance in O2 at a flow rate of 1 L/min).

  • Results: authors should provide validation data of RNA sequencing via qPCR  

>Thank you for mentioning. Herein we should note that we didn’t consider DEG singletons alone, but reconstructed experimentally verified connected network neighborhood from them (Fig. 1a), leaving only 5 from 36 DEGs unconnected to the single major network. The connected network proved joint unidirected expression alteration between groups (Fig.2a). Thus, the chance of DEGs spurious expression deviation in this instance is rather small.

The second point is that we filtered 612 DEGs for those maintaining at least fourfold expression ratio between the averages (abs(log2fold)>2).

Noteworthy, there could be low expressed genes (less than 1 FPKM), that may manifest false positives. We report that there are only 4 from 36 genes with both group average expression rate less than 1 (Table S1). The average maximum expression of 36 target DEGs 11.6 FPKM, stdev 13 FPKM. Maximum q_value was observed as 0.032 (avg q_value=0.005), average log2fold=11.5 (max log2fold=31, stdev=13.0). We believe given at least 13 genes with abs(log2fold)>10 (Table S1) we are confident in significant expression assessments, though we appreciate the experimental verification in future with more targeted studies.

Currently we are designing qPCR scheme to reassess alternatively spliced DAS genes, which are much less obvious. It might require doing higher samples size replication for more precision, or at least a second replication of experiment. For that we hope assess DAS replicability with MCAO neuroinflammation data published recently (Shishkina et al., 2021; PMID: 34944656) in our next publication, while we report rendering ‘further exploration of DAS events’ in current study. We would target first those DAS genes with Percent Spliced In (PSI) ratio close to 50% (Ptbp1, Ank2, Eogt).

Round 2

Reviewer 2 Report

Endorsed